# The perceived impact of isoniazid resistance on outcome of first-line rifampicin-throughout regimens is largely due to missed rifampicin resistance

Armand Van Deun[1][☉]*, Tom Decroo[2,3][☉], Aung Kya Jai Maug[4][‡], Mohamed Anwar Hossain[4][‡], Murid Gumusboga[1][‡], Wim Mulders[1][‡], Nimer Ortuño-Gutiérrez[5][‡], Lutgarde Lynen[2][‡], Bouke C. de Jong[1][‡], Hans L. Rieder[6][☉]

1 Biomedical Department, Mycobacteriology Unit, Institute of Tropical Medicine, Antwerp, Belgium, 2 Department of Clinical Sciences, Institute of Tropical Medicine, Antwerp, Belgium, 3 Research Foundation Flanders, Brussels, Belgium, 4 Damien Foundation Bangladesh, Dhaka, Bangladesh, 5 Damien Foundation Brussels, Brussels, Belgium, 6 Tuberculosis Consultant Services, Kirchlindach, Switzerland

☉ These authors contributed equally to this work.
‡ These authors also contributed equally to this work.
* avdeun@itg.be

**Data Availability Statement:** The data supporting the findings of this publication are retained at the Damien Foundation Belgium headquarters in

## Abstract

### Background

Meta-analyses on impact of isoniazid-resistant tuberculosis informed the World Health Organization recommendation of a levofloxacin-strengthened rifampicin-based regimen.

We estimated the effect of initial rifampicin resistance (Rr) and/or isoniazid resistance (Hr) on treatment failure or relapse. We also determined the frequency of missed initial and acquired Rr to estimate the impact of true Hr.

### Methods

Retrospective analysis of 7291 treatment episodes with known initial isoniazid and rifampicin status obtained from individual patient databases maintained by the Damien Foundation Bangladesh over 20 years. Drug susceptibility test results were confirmed by the programme's designated supra-national tuberculosis laboratory. To detect missed Rr among isolates routinely classified as Hr, *rpoB* gene sequencing was done randomly and on a sample selected for suspected missed Rr.

### Results

Initial Hr caused a large recurrence excess after the 8-month regimen for new cases (rifampicin for two months), but had little impact on rifampicin-throughout regimens: (6 months, new cases; 3.8%; OR 0.8, 95%CI:0.3,2.8; 8 months, retreatment cases: 7.3%, OR 1.8; 95% CI:1.3,2.6). Rr was missed in 7.6% of randomly selected "Hr" strains. Acquired Rr was frequent among recurrences on rifampicin-throughout regimens, particularly after the retreatment regimen (31.9%). It was higher in mono-Hr (29.3%; aOR 3.5, 95%CI:1.5,8.5) and poly-

Brussels, Belgium, and will not be made openly accessible due to ethical and privacy concerns. Data can however be made available after approval of a motivated and written request to the Damien Foundation at MTU@damiaanactie.be.

**Funding:** The authors received no specific funding for this work.

**Competing interests:** The authors have declared that no competing interests exist.

Hr (53.3%; aOR 10.2, 95%CI 4.4,23.7) than in susceptible tuberculosis, but virtually absent after the 8-month new case regimen. Comparing Bangladesh (low Rr prevalence) with a high Rr prevalence setting,true Hr corrected for missed Rr caused only 2–3 treatment failures per 1000 TB cases (of whom 27% were retreatments) in both.

## Conclusions

Our analysis reveals a non-negligible extent of misclassifying as isoniazid resistance of what is actually missed multidrug-resistant tuberculosis. Recommending for such cases a "strengthened" regimen containing a fluoroquinolone provokes a direct route to extensive resistance while offering little benefit against the minor role of true Hr tuberculosis in rifampicin-throughout first-line regimen.

## Introduction

Tuberculosis (TB) became curable with the introduction of isoniazid (INH; H).[1] Although its sterilizing activity is substantially inferior to that of e.g. rifampicin (RMP; R),[2] as evidenced by the required long duration of non-RMP-containing regimens, it has the highest early bactericidal activity among all first-line TB drugs.[3] For almost two decades WHO had recommended a standard regimen that relied on INH. Thioacetazone (Th; T) was added throughout supplemented by streptomycin (SM; S) during the first 2 months of treatment (2SHT/10HT) [notation: regimens are often abbreviated using the single-letter drug abbreviation, where a preceding number indicates the number of months in the phase whose end is indicated by a forward slash; if the drugs are given intermittently, a subscript number after the drug indicates the number of weekly dosages].[4, 5]

Because of poor results, in the early eighties standardized regimens with RMP were piloted in low-income countries by the late Dr. Styblo of the International Union against Tuberculosis and Lung Disease. In the East African settings this first "short-course" 8-month regimen (2SHRZ/6HT; PZA, Z: pyrazinamide was used (with the HIV epidemic modified to 2EHRZ/ 6HE; EMB, E: ethambutol)[6]. RMP use was limited to the intensive phase and INH given throughout treatment. Although the British Medical Research Council (BMRC) studies had already shown the high efficacy of a 6-month RMP-throughout regimen,[1] preference was given to the slightly less effective 2SHRZ/6HT regimen to reduce the risk of acquired RMP resistance during the unsupervised continuation phase to prevent then (under programme conditions) incurable MDR.[1] After an event with an INH-based regimen (designated as Category 1 regimen) indicating a high risk of isoniazid resistance (failure, relapse, return after loss from treatment), the treatment cascade envisaged a RMP-throughout retreatment regimen (2SEHRZ/1(2)EHRZ/5EHR, designated Category 2), adding RMP in the continuation phase and EMB throughout treatment.[7] Subsequently, the more efficacious 6-month regimen[8] (mainly 2EHRZ/4HR [9]) became the preferred Category 1 regimen, rendering the former 8-month regimen obsolete. [10]

A 1986 review of clinical trials showed that RMP-throughout regimens were nearly as efficacious in patients with initial resistance to INH and/or SM compared to patients with susceptible TB (3.5% versus 0.5% treatment failure).[11] These findings contrast with a 2017 meta-analysis, suggesting a higher frequency of failure or relapse among patients with H-resistant /R-susceptible TB (Hr/Rs-TB) than H- and R-susceptible TB (Hs/Rs-TB (21% versus 7% with

Category 1 and 11% versus 6% with Category 2). Acquired Rr was higher in Hr/Rs-TB than in Hs/Rs-TB (8% versus 1% with Category 1 and 3% versus 0.3% with Category 2).[12] Based on this and a second meta-analysis on treatment regimens for Hr/Rs-TB,[13] WHO revised its guidelines for Hr/Rs-TB and proposed a levofloxacin-strengthened regimen for such cases. [14] Implementation of these guidelines requires access to rapid drug susceptibility testing (DST) for all RMP, INH, and fluoroquinolone. Considering that Rr is not infrequently missed, [15] we submit that there is a tangible risk for acquired fluoroquinolone resistance.

We studied the outcome of treatment of bacteriologically positive pulmonary TB cases with various standard regimens in function of initial INH and/or RMP resistance, using data from the Damien Foundation Bangladesh (DF) project. In this project continuous drug resistance surveillance of failure and relapse cases has been a priority from the start, supplemented by periodic random population surveys.[16] We take into account that Rr at baseline can be missed, and determined its frequency in a supra-national TB reference laboratory. We also studied acquired Rr in patients with recurrent disease. Finally, we calculated the frequency of treatment failure attributable to true initial Hr (adjusted for missed Rr). We found it to be low, in line with the original BMRC clinical trials reports.

## Methods

### Design

This retrospective cohort study is based on TB patients treated in DF supported clinics in Bangladesh between 1994 and 2015.

### Setting

Since 1993 the DF Bangladesh project implements the TB treatment and care as partner in the National TB Programme (NTP). The first standardized treatment regimen for new patients was 2(3)EHRZ/6HT. In 2004, this was changed to a 6-month RMP-throughout regimen with a thrice-weekly continuation phase (2(3)EHRZ/4H$_3$R$_3$), changing to daily throughout in 2008 (2EHRZ/4HR). Between November 2014 and September 2015 patients enrolled in a trial were allocated to double-dose RMP throughout (ClinicalTrials.gov registration: NCT02153528). [17] The retreatment regimen (2SEHRZ/1(2)EHRZ/5EHR) was left unchanged.

In 2008 auramine-based LED fluorescence microscopy was phased in as replacement for brightfield Ziehl-Neelsen microscopy for acid-fast bacilli (AFB). By 2015 it was used in 75% of the over 100 field laboratories. Quality assessment of all microscopy laboratories continuously documented high-quality performance.[18]

From 2012 onward an Xpert® MTB/RIF assay network was set up, accessible to the entire population. The DF Bangladesh also runs a reference laboratory capable of culture and DST on solid media, showing excellent performance during the annual WHO DST proficiency testing rounds. From the start, the entire DF laboratory network was supported by the supra-national TB reference laboratory (SRL) of the Institute of Tropical Medicine in Antwerp, Belgium, responsible for confirmation of all DF DST, and provision of advanced techniques such as DNA sequencing. Both laboratories used phenotypic DST with the proportion method on Löwenstein-Jensen medium for INH, RMP, SM and EMB, applying standard techniques (i.e. final DST reading at 6 weeks) that remained unchanged.[19] Molecular DST was limited to RMP and INH, using Sanger sequencing of the *rpoB* gene with extended primers covering all known RMP resistance mutation-harboring regions, and Deeplex deep sequencing of the *rpoB*, *katG*, *inhA* and *inhA* promoter genes.

Between 1995 and 2015, Rr and Hr prevalence remained stable in new patients, around 1% and 5% respectively. With the change to the 6-month Category 1 regimen, Rr rose from 8% to 50% while Hr fell from 33% to 8% among retreatment patients.[16, 20–22]

## Data collection and structuring of the study databases

We used routine patient, reference laboratory, and specific study databases to link information on pre-existing drug resistance to treatment outcome. Individual first-line patient databases containing most information from routine treatment cards were maintained systematically until the end of 2013. Each patient treatment episode was given a unique identifier. The database was periodically updated for bacteriologically positive recurrence subsequent to successful treatment completion. If a recurrence was declared within two years after cure or completion it was defined as relapse. Culture and/or DST results were retrieved from the DF Bangladesh reference laboratory database on continuous drug resistance surveillance among presumed drug-resistant cases (retreatments, late converters, new cases contacts of Rr TB, but almost no new cases), as described.[16] On average, 80% to 90% of relapse and failure cases from all project clinics had recurrence sputum specimens referred. Drug resistance data originating from *ad hoc* studies were used as well. Phenotypic initial resistance profiles of randomly selected new cases came from two drug resistance surveys (DRS) conducted in six randomly selected sentinel clinics in 1995 and 2001.[21] Molecular DST from alcohol-preserved sputa was used for two DRS cohorts. Each of these comprised about 500 successively recorded new patients in 2005, and again in 2010 from the same sentinel clinics.[23]. The same was done for a 2015 study cohort of randomly selected new cases from a clinical study on double-dose RMP with about 950 patients, again from these sentinel clinics (NCT02153528).

Databases were conceived and handled in EpiData Entry 3.1 (EpiData Association, Odense, Denmark). Single data entry of the close to 300,000 records was combined with regular checks and corrections by layers of supervisors. Database cleaning, in particular checking registered outcomes against recorded smear results and consistent recurrence definitions was done for all DRS and study databases prior to analysis, while the reference laboratory database was cleaned annually.

The information recorded in the treatment as well as reference laboratory databases included a unique patient-treatment-episode identifier, as well as a unique lifetime patient-identifier, the latter issued only to patients presumed or confirmed to have drug resistance. The various identifiers allowed correctly linking laboratory and patient information. Twenty-three linked records were excluded because the link was clearly erroneous when comparing patient initials, age, and sex from the two parent databases.

DST results from different samples tested for the same treatment episode and indication were condensed into a single record in which any positive culture or a resistant DST result overrode other values.

DST results obtained 60 or more days after treatment initiation were excluded, since drug resistance might already have been amplified from baseline by that time. Linked records with a resistance profile obtained more than 90 days before treatment start were also excluded, as possibly belonging to a previous treatment episode.

A resistant RMP result from any of the reference laboratories and by any technique overrode all other results for the same condensed record. Missed RMP resistance shown by *rpoB* sequencing had been added to the database for suspected strains selected for previous studies, and if present, this result was used for analysis. Comparing exclusively phenotypic DST-based profiles with those including an eventually present overriding molecular Rr result, corrected missed Rr amounted to 4% of all Rr. Over the last 20 years, the Antwerp SRL has been among

those with the highest sensitivity for RMP resistance detection in the WHO annual rounds of DST EQA for the SRL global network (unpublished data from annual reports received by our SRL).

## Definition of variables

Initial Hr included mono-resistance (mono-Hr) and poly-resistance (poly-Hr; defined here as resistance to isoniazid plus EMB (HE) and/or SM (HES or HS), while susceptible to RMP). Initial Hr was assessed in the entire study population.

Rr in baseline samples was defined as missed Rr if routine phenotypic DST showed Rs-TB but Sanger *rpoB* sequencing done *a posteriori* at the SRL showed Rr [24]. A random selection of about 200 strains registered as Hr/Rs-TB was added to the data set of patients with a high risk of Rr-TB (retreatment, late conversion, new contacts of cases with Rr TB), for testing on missed Rr.

Rr in recurrence samples was defined as acquired Rr if the baseline sample for the same episode showed Rs-TB. Acquired Rr was assessed in all patients with initial Rs-TB for whom a DST results for a recurrence sample were available.

Successful outcomes were cure or treatment completion without bacteriological proof of relapse during the first two years after treatment. Death, lost to follow-up, treatment failure or relapse were defined as programmatically adverse outcomes. Bacteriologically adverse outcomes were AFB smear microscopy-based treatment failure or relapse. NTP and WHO guidelines and definitions were strictly followed. The smear microscopy cut-off for diagnosis as well as declaration of failure or relapse was at least 1 acid-fast bacillus (AFB) in at least 1 smear.

## Data analysis

Univariable and multivariable logistic regression was used to identify factors, such as initial resistance profile and treatment regimen, that might be associated with treatment outcomes, programmatic or bacteriological, missed Rr, and acquired Rr. We show (adjusted) odds ratios (OR) and 95% confidence intervals (CI).

## Comparison with a high MDR prevalence setting

The DF Bangladesh environment is a low MDR-TB setting. To evaluate our calculations for a corrected Hr frequency, we used a data set from Georgia with a high MDR prevalence, applying our methodology to it to have an external comparison using our approach to settings with a different epidemiology. We selected Georgia's published data set because the required data were readily available, [25] and because the methods and accuracy of results for RMP between our study SRL and the Georgia National Reference Laboratory were almost identical.

### Ethics

The study protocol was approved by the Institutional Review Board of the ITM, Antwerp, Belgium; since it concerned anonymous routinely collected data, the requirement for informed consent was waived.

## Results

Fig 1 shows the study population. The subpopulations in which missed (N = 382) or acquired (N = 242) Rr were studied, were part of the main effectiveness population of 7291 cases. Patients who were chosen for DST had been selected randomly among successively registered cases or systematically among incident retreatment cases, with high risk of drug-resistant TB.

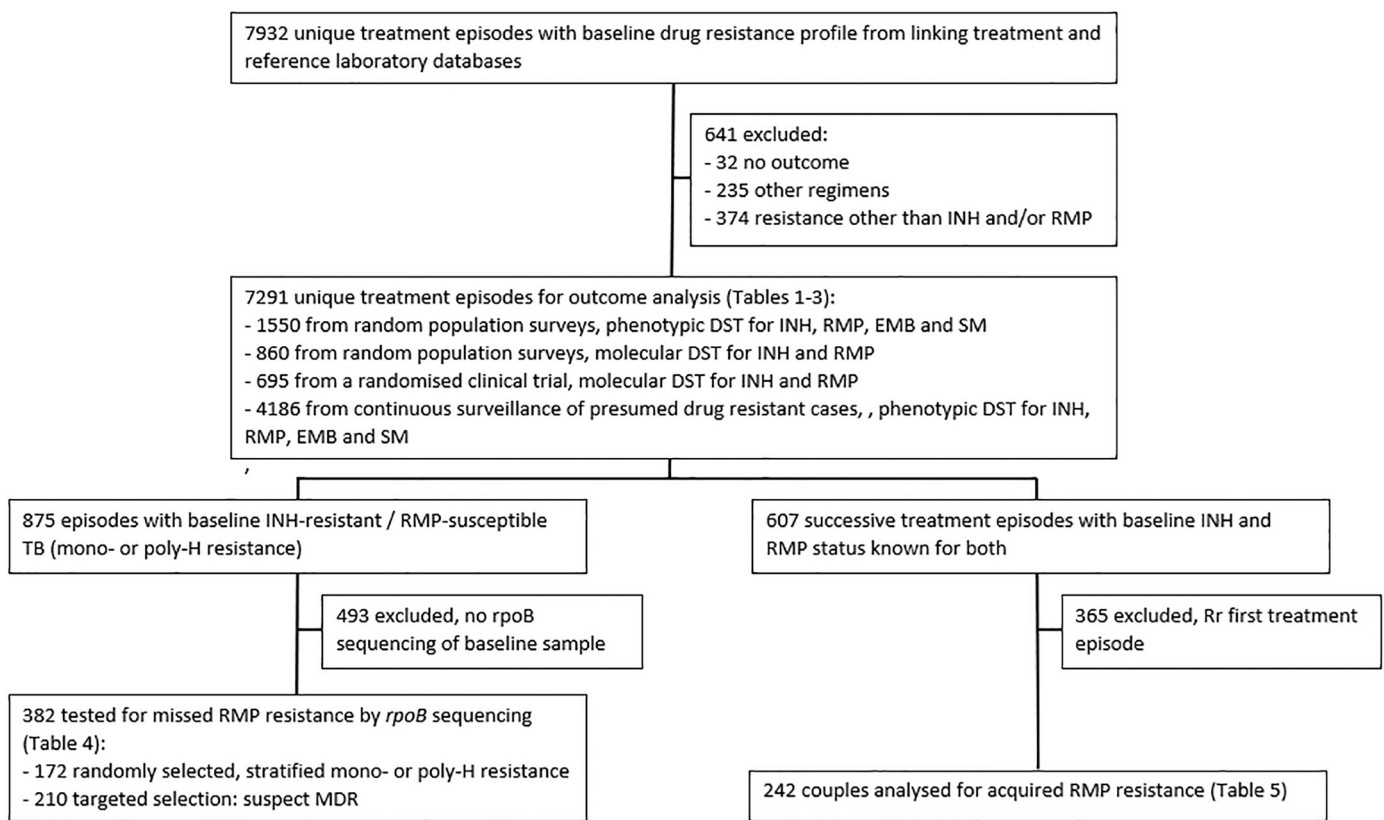

**Fig 1. Flowchart showing the analysed populations.** INH: isoniazid; RMP: rifampicin; EMB: ethambutol; SM: streptomycin; DST: drug susceptibility testing; TB: tuberculosis.

New, previously untreated cases were not systematically tested and not included in this database, with the exception of symptomatic MDR-TB contacts (N = 47).

Table 1 shows programmatical and bacteriological effectiveness for 7291 treatment episodes, stratified by treatment regimens and by initial resistance profile. Patients with Hs/Rs-TB treated with 2(3)EHRZ/6HT, the 6-month RMP regimens 2(3)EHRZ/4H₃R₃, 2(3)EHRZ/4HR and 2EHR$^h$Z/4HR$^h$ and Category 2 were reported with 96.0%, 95.2%, 95.2%, 95.8% and 95.6% bacteriological success, respectively.

### Effect of treatment regimen, by initial resistance pattern

Among patients with Hs/Rs-TB, treatment with Category 2 more likely resulted in a programmatically adverse outcome (vs. 6-month RMP-throughout regimen; OR 1.8; 95%CI 1.5–2.1; Table 2). However, treatment regimen was not associated with having a bacteriologically adverse outcome (failure or relapse).

Among patients with Rr-TB bacteriological adverse outcomes were frequent, ranging between 52.9% and 61.9%, and there were no significant differences between regimens.

Among patients with Hr/Rs-TB, 27.5% experienced either failure or relapse when treated with 2(3)EHRZ/6HT. Table 1 shows that these proportions were far higher with poly-Hr (42.4%, 14/33; mostly HS resistant) than with mono-Hr (17.5%, 10/57), even though the regimen did not contain streptomycin. Table 2 shows that treatment with 2(3)EHRZ/6HT

**Table 1. Treatment outcomes of 7291 patients on first-line TB treatment regimens in Bangladesh, by initial resistance to rifampicin and isoniazid.**

| | Total | Failure | | Relapse | | Died | | LTFU | | Relapse-free success | Programmatical effectiveness § | Bacteriological effectiveness # |
|---|---|---|---|---|---|---|---|---|---|---|---|---|
| | N | N | % | N | % | N | % | N | % | N | % | % |
| **2(3)EHRZ/6HT** | | | | | | | | | | | | |
| Total | 1471 | 52 | 3.5 | 30 | 2 | 68 | 4.6 | 85 | 5.8 | 1236 | 84.0 | 93.8 |
| Susceptible to INH and RMP | 1349 | 31 | 2.3 | 17 | 1.3 | 62 | 46 | 77 | 5.7 | 1162 | 86.1 | 96.0 |
| RMP resistance † | 20 | 6 | 30.0 | 3 | 15 | 1 | 5.0 | 2 | 10.0 | 8 | 40.0 | 47.1 |
| INH mono-resistance | 66 | 7 | 10.6 | 3 | 4.5 | 5 | 7.6 | 4 | 6.1 | 47 | 71.2 | 82.5 |
| INH poly-resistance ‡ | 35 | 7 | 20.0 | 7 | 20.0 | 0 | 0.0 | 2 | 5.7 | 19 | 54.3 | 57.6 |
| INH resistance, unknown whether mono- or poly-resistance | 1 | 1 | 100.0 | 0 | 0.0 | 0 | 0.0 | 0 | 0.0 | 0 | 0.0 | 0.0 |
| **2(3)EHRZ/4H₃R₃** | | | | | | | | | | | | |
| Total | 358 | 10 | 2.8 | 10 | 2.8 | 14 | 3.9 | 24 | 6.7 | 300 | 83.8 | 93.8 |
| Susceptible to INH and RMP | 331 | 5 | 1.5 | 9 | 2.7 | 13 | 3.9 | 24 | 7.3 | 280 | 84.6 | 95.2 |
| RMP resistance † | 6 | 3 | 50.0 | 1 | 16.7 | 1 | 16.7 | 0 | 0.0 | 1 | 16.7 | 20.0 |
| INH mono-resistance | 6 | 0 | 0.0 | 0 | 0.0 | 0 | 0.0 | 0 | 0.0 | 6 | 100.0 | 100.0 |
| INH poly-resistance ‡ | 2 | 2 | 100.0 | 0 | 0.0 | 0 | 0.0 | 0 | 0.0 | 0 | 0.0 | 0.0 |
| INH resistance, unknown whether mono- or poly-resistance | 13 | 0 | 0.0 | 0 | 0.0 | 0 | 0.0 | 0 | 0.0 | 13 | 100.0 | 100.0 |
| **2(3)EHRZ/4HR** | | | | | | | | | | | | |
| Total | 1018 | 43 | 4.2 | 11 | 1.1 | 31 | 3 | 28 | 2.8 | 905 | 88.9 | 94.4 |
| Susceptible to INH and RMP | 928 | 34 | 3.7 | 8 | 0.9 | 26 | 2.8 | 27 | 2.9 | 833 | 89.8 | 95.2 |
| RMP resistance † | 23 | 8 | 34.8 | 2 | 8.7 | 3 | 13.0 | 0 | 0.0 | 10 | 43.5 | 50.0 |
| INH mono-resistance | 9 | 1 | 11.1 | 1 | 11.1 | 1 | 11.1 | 1 | 11.1 | 5 | 55.6 | 71.4 |
| INH poly-resistance ‡ | 8 | 0 | 0.0 | 0 | 0.0 | 0 | 0.0 | 0 | 0.0 | 8 | 100.0 | 100.0 |
| INH resistance, unknown whether mono- or poly-resistance | 50 | 0 | 0.0 | 0 | 0.0 | 1 | 2.0 | 0 | 0.0 | 49 | 98.0 | 100.0 |
| **2EHR^hZ/4HR^h** | | | | | | | | | | | | |
| Total | 391 | 11 | 2.8 | 4 | 1.0 | 5 | 1.3 | 10 | 2.6 | 361 | 92.3 | 96.0 |
| Susceptible to INH and RMP | 370 | 11 | 3 | 4 | 1.1 | 5 | 1.4 | 10 | 2.7 | 340 | 91.9 | 95.8 |
| RMP resistance † | 1 | 0 | 0.0 | 0 | 0.0 | 0 | 0.0 | 0 | 0.0 | 1 | 100.0 | 100.0 |
| INH resistance, unknown whether mono- or poly-resistance | 20 | 0 | 0.0 | 0 | 0.0 | 0 | 0.0 | 0 | 0.0 | 20 | 100.0 | 100.0 |
| **2SEHRZ/1(2)EHRZ/5EHR** | | | | | | | | | | | | |
| Total | 4053 | 288 | 7.1 | 138 | 3.4 | 246 | 6.1 | 361 | 8.9 | 3020 | 74.5 | 87.6 |
| Susceptible to INH and RMP | 2759 | 47 | 1.7 | 57 | 2.1 | 153 | 5.5 | 232 | 8.4 | 2270 | 82.3 | 95.6 |
| RMP resistance † | 571 | 204 | 35.7 | 70 | 12.3 | 65 | 11.4 | 63 | 11.0 | 169 | 29.6 | 38.1 |
| INH mono-resistance | 387 | 13 | 3.4 | 8 | 2.1 | 12 | 3.1 | 35 | 9.0 | 319 | 82.4 | 93.8 |
| INH poly-resistance ‡ | 332 | 23 | 6.9 | 3 | 0.9 | 16 | 4.8 | 30 | 9.0 | 260 | 78.3 | 90.9 |
| INH resistance, unknown for whether mono- or poly-resistance | 4 | 1 | 25.0 | 0 | 0.0 | 0 | 0 | 1 | 25.0 | 2 | 50.0 | 66.7 |

INH or H: isoniazid; RMP or R: rifampicin; E: ethambutol; S: streptomycin; T: thioacetazone; Z: pyrazinamide; h superscript: high-dose; LTFU: lost to follow-up

Preceding numbers in regimens are months of treatment; subscript numbers indicate intermittent number of doses of preceding drug per week

† Regardless of resistance to INH

‡ INH poly-resistance: includes HS-resistant TB, HE-resistant TB, and HES-resistant TB (H = isoniazid; S = streptomycin; E = ethambutol). The vast majority is HS-resistant TB

§ Programmatical effectiveness: cure or completion against the same plus failure and relapse, death or LTFU

# Bacteriological effectiveness: cure or completion against the same plus failure and relapse

**Table 2. Effect of treatment regimen on outcome, by initial resistance profile, in 7291 patients.**

| | | Programmatically adverse outcome† | | | | | Bacteriologically adverse outcome‡ | | | |
|---|---|---|---|---|---|---|---|---|---|---|
| | Total | n | % | OR | [95%CI] | Total | n | % | OR | [95%CI] |
| **Initially susceptible to RMP and INH** | | | | | | | | | | |
| 6-month RMP-throughout regimen § | 1629 | 176 | 10.8 | 1 | | 1524 | 71 | 4.7 | 1 | |
| 2(3)EHRZ/6HT | 1349 | 187 | 13.9 | 1.3* | [1.1,1.7] | 1210 | 48 | 4.0 | 0.8 | [0.6,1.2] |
| 2SEHRZ/1(2)EHRZ/5EHR | 2759 | 489 | 17.7 | 1.8*** | [1.5,2.1] | 2374 | 104 | 4.4 | 0.9 | [0.7,1.3] |
| **Initially resistant to RMP** | | | | | | | | | | |
| 6-month RMP-throughout regimen § | 30 | 18 | 60.0 | 1 | | 26 | 14 | 53.8 | 1 | |
| 2(3)EHRZ/6HT | 20 | 12 | 60.0 | 1 | [0.3,3.2] | 17 | 9 | 52.9 | 1 | [0.3,3.3] |
| 2SEHRZ/1(2)EHRZ/5EHRE | 571 | 402 | 70.4 | 1.6 | [0.7,3.4] | 443 | 274 | 61.9 | 1.4 | [0.6,3.1] |
| **RMP susceptible, INH resistant** | | | | | | | | | | |
| 6-month RMP-throughout regimen § | 108 | 7 | 6.5 | 1 | | 105 | 4 | 3.8 | 1 | |
| 2(3)EHRZ/6HT | 102 | 36 | 35.3 | 7.9*** | [3.3,18.7] | 91 | 25 | 27.5 | 9.6*** | [3.2,28.7] |
| 2SEHRZ/1(2)EHRZ/5EHR | 723 | 142 | 19.6 | 3.5** | [1.6,7.8] | 629 | 48 | 7.6 | 2.1 | [0.7,5.9] |

INH or H: isoniazid; RMP or R: rifampicin; E: ethambutol; S: streptomycin; T: thioacetazone; Z: pyrazinamide; OR: odds ratio

Preceding numbers in regimens are months of treatment; subscript numbers indicate intermittent number of doses of preceding drug per week

Level of significance:

** p<0.01;

*** p<0.001

† programmatically adverse: failure, relapse, death or LTFU

‡ bacteriologically adverse: failure or relapse

§ either 2(3)EHRZ/4H$_3$R$_3$, 2(3)EHRZ/4HR, or 2EHR$^h$Z/4HR$^h$

compared to a 6-month RMP regimen more likely resulted in a programmatically (OR 7.9; 95%CI:3.3–18.7) or bacteriologically (OR 9.6; 95%CI 3.2–28.7) adverse outcome. Of Hr/Rs-TB patients treated with Category 2, 19.6% and 7.6% had a programmatically and bacteriologically adverse outcome, respectively. Category 2 compared to the 6-month RMP regimen was more likely to result in a programmatically (OR 3.5; 95%CI 1.6–7.8) but not bacteriologically (OR 2.1; 95%CI 0.7–5.9) adverse outcome.

## Effect of initial resistance pattern, by regimen

Among patients treated with the 6-month RMP-throughout regimen, initial Hr had no statistically significant effect on outcome if the strain was susceptible to RMP (Table 3). This was not the case for other regimens. Among patients treated with 2(3)EHRZ/6HT initial Hr vs. Hs/Rs-TB was strongly associated with a programmatically (OR 3.4; 95%CI:2.2–5.2) and bacteriologically (OR 9.2; 95%CI:5.3–15.8) adverse treatment outcome. Among patients treated with Category 2 for Hr/Rs-TB, initial Hr vs. Hs/Rs-TB was associated with having a bacteriologically adverse outcome (aOR 1.8, 95%CI: 1.3–2.6).

## Missed initial Rr in patients with Hr-TB

In 13.6% (52/382) of patients with Hr/Rs TB on phenotypic DST, genotypic DST showed Rr *a posteriori* (Table 4). Initial Rr was missed in 4.5% (1/22) of new patients tested. Initial Rr was missed more frequently after a bacteriologically unsuccessful Category 2 than after unsuccessful 6-month RMP treatment (36.1% vs 7.4%; aOR 6.8; 95%CI 3.3–13.8). Initial Rr was also missed more frequently with poly-Hr than mono-Hr (15.9% vs. 10.9%), without reaching statistical significance.

**Table 3. Effect of initial resistance on outcomes, by treatment regimen, in 7291 patients.**

| | | Programmatically adverse outcome[†] | | | | Bacteriologically adverse outcome[‡] | | | | |
|---|---|---|---|---|---|---|---|---|---|---|
| | Total | n | % | OR | [95%CI] | Total | n | % | OR | [95%CI] |
| **6-month RMP-throughout regimen §** | | | | | | | | | | |
| Susceptible to INH and RMP | 1629 | 176 | 10.8 | 1 | | 1524 | 71 | 4.7 | 1 | |
| RMP resistant | 30 | 18 | 60.0 | 12.4*** | [5.9,26.1] | 26 | 14 | 53.8 | 23.9*** | [10.7,53.5] |
| RMP susceptible, INH resistant | 108 | 7 | 6.5 | 0.6 | [0.3,1.3] | 105 | 4 | 3.8 | 0.8 | [0.3,2.3] |
| **2(3)EHRZ/6HT** | | | | | | | | | | |
| Susceptible to INH and RMP | 1349 | 187 | 13.9 | 1 | | 1210 | 48 | 4.0 | 1 | |
| RMP resistant | 20 | 12 | 60.0 | 9.3*** | [3.8,23.1] | 17 | 9 | 52.9 | 27.2*** | [10.1,73.7] |
| RMP susceptible, INH resistant | 102 | 36 | 35.3 | 3.4*** | [2.2,5.2] | 91 | 25 | 27.5 | 9.2*** | [5.3,15.8] |
| **2SEHRZ/1(2)EHRZ/5EHR** | | | | | | | | | | |
| Susceptible to INH and RMP | 2759 | 489 | 17.7 | 1 | | 2374 | 104 | 4.4 | 1 | |
| RMP resistant | 571 | 402 | 70.4 | 11.0*** | [9.0,13.5] | 443 | 274 | 61.9 | 35.4*** | [26.9,46.6] |
| RMP susceptible, INH resistant | 723 | 142 | 19.6 | 1.1 | [0.9,1.4] | 629 | 48 | 7.6 | 1.8** | [1.3,2.6] |

INH or H: isoniazid; RMP or R: rifampicin; E: ethambutol; S: streptomycin; T: thioacetazone; Z: pyrazinamide; OR: odds ratio

Preceding numbers in regimens are months of treatment; subscript numbers indicate intermittent number of doses of preceding drug per week

Level of significance:

** p<0.01;

*** p<0.001

[†] programmatically adverse: either failure, relapse, death or LTFU

[‡] bacteriologically adverse: either failure or relapse

§ either 2(3)EHRZ/4H$_3$R$_3$, 2(3)EHRZ/4HR, or 2EHR$^h$Z/4HR$^h$

**Table 4. Missed initial rifampicin resistance, among 382 patients with initial rifampicin-susceptible/isoniazid-resistant TB on phenotypic DST.**

| | Total | Initial Rs confirmed | | Initial Rr missed | | | | | |
|---|---|---|---|---|---|---|---|---|---|
| | N | N | % | N | % | OR | [95%CI] | aOR | [95%CI] |
| **All †** | 382 | 330 | 86.4 | 52 | 13.6 | NA | | NA | |
| **INH resistance profile** | 382 | | | | | | | | |
| INH mono-resistance | 174 | 155 | 89.1 | 19 | 10.9 | 1 | | 1 | |
| INH poly-resistance | 208 | 175 | 84.1 | 33 | 15.9 | 1.5 | [0.8,2.8] | 1.5 | [0.8,2.8] |
| **Criteria for referral for sequencing** | | | | | | | | | |
| Random selection | 210 | 194 | 92.4 | 16 | 7.6 | 1 | | 1 | |
| Presumptive Rr-TB $ | 172 | 136 | 79.1 | 36 | 20.9 | 3.2*** | [1.7,6.0] | 3.2*** | [1.6,6.2] |
| **Treatment history** | | | | | | | | | |
| First TB episode | 22 | 21 | 95.5 | 1 | 4.5 | 0.6 | [0.1,4.7] | 1.1 | [0.1,9.5] |
| Failure or relapse of a first treatment | 231 | 214 | 92.6 | 17 | 7.4 | 1 | [1.0,1.0] | 1 | |
| Failure or relapse of a retreatment | 72 | 46 | 63.9 | 26 | 36.1 | 7.1*** | [3.6,14.2] | 6.8*** | [3.3,13.8] |
| Other ‡ | 57 | 49 | 86 | 8 | 14 | 2.1 | [0.8,5.0] | 2.3 | [0.9,5.7] |

INH: Isoniazid; Rr-TB: rifampicin-resistant TB; Rs: rifampicin susceptible; N: number; NA: not applicable; OR: odds ratio; aOR: adjusted odds ratio

[†] Multiple samples per episode possible

[‡] Other: includes unknown antecedents, non-conform regimens and all lost to follow-up

$, highly unusual resistance profiles, Rs failures of RMP-throughout regimens, new cases contacts of Rr TB

Level of significance:

*** p<0.001

**Table 5. Predictors of acquired rifampicin resistance in 242 treatment failure and relapse patients with initially rifampicin-susceptible TB.**

|  | Total with failure or relapse | Acquired Rr | | OR † | [95%CI] | aOR † | [95%CI] |
|---|---|---|---|---|---|---|---|
|  |  | N | % |  |  |  |  |
| **Total** | 242 | 56 | 23.1 | NA |  | NA |  |
| **Regimen during which Rr was acquired** |  |  |  |  |  |  |  |
| 2(3)EHRZ/6HT | 68 | 1 | 1.5 | 1 |  | 1 |  |
| 6-month RMP-throughout regimen ‡ | 14 | 4 | 28.6 | 19.3** | [2.7,136.8] | 29.6** | [3.6,241.5] |
| 2SEHRZ/1(2)EHRZ/5EHR | 160 | 51 | 31.9 | 21.2*** | [4.1,110.4] | 26.8*** | [4.9,146.1] |
| **Resistance pattern before Rr was acquired §** |  |  |  |  |  |  |  |
| Susceptible to INH and RMP | 155 | 19 | 12.3 | 1 |  | 1 |  |
| INH mono-resistance | 41 | 12 | 29.3 | 3.0** | [1.3,6.7] | 3.5** | [1.5,8.5] |
| INH poly-resistance | 45 | 24 | 53.3 | 8.0*** | [3.8,16.9] | 10.2*** | [4.4,23.7] |

INH or H: isoniazid; RMP or R: rifampicin; E: ethambutol; S: streptomycin; T: thioacetazone; Z: pyrazinamide; N: number; OR: odds ratio, aOR: adjusted odds ratio; Rr: rifampicin resistance

Preceding numbers in regimens are months of treatment; subscript numbers indicate intermittent number of doses of preceding drug per week

† To account for the effect of rare events Firth logistic regression method was used

‡ either 2(3)EHRZ/4H$_3$R$_3$, 2(3)EHRZ/4HR, or 2EHR$^h$Z/4HR$^h$

§ results not shown for one patient with INH resistance, unknown for EMB and SM

Level of significance:

** p<0.01;

*** p<0.001

## Acquired Rr in patients with recurrence

Of 242 Rs-TB patients with a bacteriologically unsuccessful first-line treatment 23.1% acquired Rr (Table 5). Acquired Rr was found far more frequently after bacteriologically unsuccessful treatment with a 6-month RMP regimen (28.6%; aOR 29.6; 95%CI 3.6–241.5) or Category 2 (31.9%; aOR 26.8; 95%CI:4.9–146.1) than after unsuccessful treatment with 2(3)EHRZ/6HT (1.5%: of 68, only one (initially poly-Hr) patient had acquired Rr).

Mono-Hr vs. Hs/Rs-TB (aOR 3.5; 95%CI 1.5–8.5) and particularly poly-Hr (aOR 10.2; 95% CI: 4.4–23.7) were associated with acquired Rr after recurrence. All four with acquired Rr after 6M-RMP were MDR contacts, 3 out of the 5 with poly-Hr (for a total of 11 Hr contacts on this regimen): one Hs/Rs-TB patient acquired mono-Rr and relapsed and three with initially poly-Hr/Rs TB (1 HS, 2 EHSZ resistant) on deep sequencing acquired Rr after treatment failure.

## Comparison with data from a high MDR setting

We used the published data from Georgia as an external dataset to determine comparatively the impact of missed RMP resistance in a high-MDR setting.

Table 6 shows the effect of missed Rr among smear-positive pulmonary TB patients with Hr-TB treated with a first-line regimen in Georgia (higher Rr prevalence) and Bangladesh (lower Rr prevalence). Assuming 8% of missed Rr among "Hr" in both settings, of which half had already been corrected for the Bangladesh outcome analyses, we estimate that 72.2% (6.5/ 9.0) of treatment failures were due to missed initial Rr in Georgia, against 51.3% (2/3.9) in Bangladesh. In both settings, true Hr/Rs-TB would cause 2–3 failures per 1000 TB patients, new Category 1 and retreatment Category 2 combined at a 730/270 ratio (as in the Georgia publication).

**Table 6. Effect of missed rifampicin resistance on treatment outcomes in patients with isoniazid resistance treated with a first-line regimen, in a setting with high (21%, Georgia) and low (13%, Bangladesh) rifampicin resistance prevalence (combined, new and retreatment cases together at the same proportions).**

| Initial RMP/INH resistance pattern | Bangladesh | Georgia |
|---|---|---|
| | | No per 1000 cases |
| Hs/Rs-TB | 802.0 | 690.0 |
| Failure in Hs/Rs-TB (2.2%) | 18.0 | 15.0 |
| Rr-TB | 130.0 | 210 |
| Failure in Rr-TB (if treated with a RMP-throughout regimen; 36%) | 47.0 | 75.6 |
| Hr/Rs-TB | 62.3 | 82.0 |
| Failure in Hr/Rs-TB (3.4%) | 1.9 | 2.5 |
| Hr-TB with missed Rr | 5.7 | 18.0 |
| Failure in Hr-TB with missed Rr (36%) | 2.0 | 6.5 |

Hr: Isoniazid-resistant; Hs: isoniazid-susceptible Rr: rifampicin-resistant; Rs: rifampicin-susceptible

## Discussion

In our setting, the 6-month RMP Category 1 regimen was equally successful for susceptible and Hr/Rs-TB patients. In contrast, this was not the case with the predecessor Category 1 8-month 2(3)EHRZ/6HT regimen where INH was the key drug during the continuation phase. The frequency of the smear microscopy-defined bacteriologically adverse outcomes was 3.8% in patients with Hr/Rs-TB treated with the 6-month RMP-throughout regimen and 7.6% in patients treated with the Category 2 RMP-throughout retreatment regimen. Our study is observational, is from a single setting, and covers a limited number of patients with Hr/Rs-TB treated with a 6-month RMP-throughout regimen, the failure frequency is very close to the 3.5% reported from the original BMRC clinical trials. [1]

### Discrepancy with the meta-analysis on the effect of Hr

Our findings show better outcomes than those reported by a 2017 meta-analysis, which had shown 21% and 11% bacteriologically adverse outcomes after a 6-month RMP-based Category 1 and Category 2 regimen for Hr/Rs-TB respectively.[12] It seems noteworthy that a very small fraction of studies included in the meta-analysis dominated the findings. For instance, for those categorised as standard 6-month RMP regimen (WHO new), 6/25 studies comprise half (673/1269) of the Hr/Rs-TB cases and three quarters (172/229) of the recurrences.[12] Moreover, inclusion of some of these and other large studies may have caused additional and serious bias. One study, which was categorized as evaluating a 6-month RMP-throughout regimen, evaluated actually 2SHRZ/6HE, similar to the 2EHRZ/6HT regimen that we found little effective for patients with Hr-TB [1], [26]. Four clinical studies from India reported up to 65% recurrence for retreatment regimens with twice- or thrice-weekly dosing throughout.[27, 28], [29] The evidence for daily treatment throughout (and notably during the intensive phase) is established [30], while 6 month regimens, fully intermittent at the frequencies used in these studies, have a strongly increased risk for relapse.[31] Other large studies were conducted in settings with a high Rr prevalence, where missed initial Rr may explain excess treatment failure. [29, 32, 33] In Tomsk, the large majority of patients with Hr-TB had poly-resistance (even EHS-resistance, generally a rare profile),[33] putting them at high risk of acquiring Rr, if initial Rr had not been missed in the first place. A large multi-country study conducted in high Rr prevalence settings had shown 4% (17/392) treatment failure in Hr-TB patients treated with

6-month RMP-throughout regimens. However, in line with our findings this rose to 24% (21/88) with Category 2 treatment, [34] although bacteriologically a Category 2 regimen cannot be weaker than the 6-month RMP regimen. More missed and/or acquired Rr for Category 2 is the most likely explanation. This likely also applies to our own, only partly corrected and less extreme findings. Bacteriological outcome definitions of the meta-analysis were not harmonized. Relapse was not always reported, even in the largest studies such as the one reported by Espinal and colleagues. [34] Some studies used culture-defined failure and relapse, other studies used smear microscopy-defined outcomes. Finally, in stark contrast to our study, the variation in regimens and study populations and the lack of uniform outcome definitions all contributed to the large heterogeneity shown in the meta-analysis to an extent that strongly suggests inconsistency of findings,[35] which should preclude generalization into one-directional guidelines.

## Rr missed by phenotypic and genotypic DST is not rare

Rr caused by mutations with lower level resistance and/or fitness loss frequently escapes phenotypic DST, particularly with rapid methods such as the MGIT system.[23, 36] These mutations have a similar detrimental impact on bacteriological outcome as the common *rpoB* mutations.[23]. Molecular tests do not have this problem but they miss a few Rr mutations outside the Rr-determining region.[37–39] Before lot 00057, the Hain Genotype line probe assay (LPA) missed the not so rare 533Pro mutation (Hain Lifesciences communication, 2013)[40], conferring low-level resistance that is generally also missed by MGIT960 DST. The high frequency of failure with the 6-month RMP-based regimen in Hr-Rs cases determined with the Hain LPA in an Indian study incorporated into the meta-analysis might thus have been caused by missed Rr.[29, 41] However, the dominant reason is likely heteroresistance. This is easily missed by molecular DST when both susceptible and mutant *M. tuberculosis* populations are present.[42] Xpert® MTB/RIF as well as Xpert® MTB/RIF Ultra validations have shown that heteroresistance is poorly detected.[43] Phenotypic tests, by design the most sensitive for heteroresistance, may also miss it because of overgrowth by the fitter susceptible wild-type strain during primary culture and DST incubation.[44]

The proportion of missed Rr among those with treatment failure is higher when the prevalence of MDR-TB is higher, because of the similar frequency of easily missed low-level resistance types (as a group) among all *rpoB* mutations.[37] Missed Rr may explain the relatively high frequency of adverse bacteriological outcomes in settings such as Georgia, with 4% treatment failures and 5% switched to MDR treatment in patients with initially Hr/Rs-TB treated with a dedicated 9-month RZE regimen.[25] According to our calculations, and after correction for missed Rr, the proportion with true Hr-TB treatment failure after treatment with 6M-RMP was very similar, and at 2–3 per 1000 treated TB cases (including 27% retreatments), similarly low for the two settings with a respectively low and high MDR-TB prevalence.

## Implications for treatment regimens

Missed Rr can have major implications. Based on the 2018 meta-analysis,[12, 13] WHO guidelines now recommend INH and fluoroquinolone DST in previously treated patients with proven Rs-TB, and the use of a levofloxacin-strengthened regimen 6E(H)RZ -Lfx in patients with Hr/Rs-TB.[14] However, if Rr is indeed missed in a substantial proportion of patients with Hr-TB, such patients with actual MDR-TB are left only with ethambutol to prevent acquisition of fluoroquinolone resistance (pyrazinamide not being active against the bulk of actively dividing bacilli in alkaline environment). Extensive drug resistance could even become a frequent outcome if retreatment patients, with their higher proportion of missed as well as

acquired Rr, are systematically allocated to this regimen. In view of the inability of most NTPs to provide timely isoniazid susceptibility results, such a strategy might become quite popular.

A safer and more feasible alternative than testing for Hr and using the WHO recommended levofloxacin-strengthened regimen in those with Hr-TB might be the use of the 6-month EHRZ regimen giving INH throughout (6EHRZ, rather than 6E(H)RZ)[45] in all retreatment patients with Rs-TB, regardless of an INH DST result.[13] According to our calculations above, the longer 9ERZ regimen without INH used in Georgia and meant to overcome Hr-TB seems to result more frequently in failure attributable to true Hr-TB than the 6-month RMP-based (2EHRZ/4HR) regimen in Bangladesh. From the data provided for 6–9 months E(H)RZ regimens and the corresponding publications, it appears that 7/423 (2%) versus 90/927 (10%) recurrences occurred in those treated with or without INH respectively. Our findings showing 70–80% treatment success in patients with mono-Hr TB treated with the 2EHRZ/6HT regimen, relying virtually solely on INH in the continuation phase, also suggest remaining activity of INH despite *in vitro* resistance. Regimens with up to quadruple-dose INH and/or RMP for retreatment should be tested. Such high doses are more sterilizing for true Rs-TB. They maximize the residual effect of INH in Hr/Rs-TB and possibly also that of RMP in case of the more frequently missed low-level resistance mutations.[46, 47]

## Acquired Rr in patients with recurrence

Acquired Rr is thought to be relatively rare after unsuccessful treatment with the standard 6-month RMP-throughout regimen. However, among recurrent cases after Category 1 and Category 2 treatment acquired Rr was far higher than after the 8-month 2(3)HRZE/6HT regimen (one third of recurrences after Category 2 vs. 1.5% of recurrences after 2(3)HRZE/6HT). This testifies that it was a correct judgment by NTPs to give preferenceto the 8-month regimen at that time when RMP resistance was incurable outside affluent countries. Besides RMP-throughout treatment, other predictors of acquired Rr were initial mono-Hr and particularly poly-Hr.

Strikingly, all four patients documented with acquired Rr after the 6-month RMP-based regimen were contacts of MDR cases. Two had initial EHSZ resistance, *de facto* leading to RMP monotherapy. But it is also not unreasonable to assume that these contacts testing Rs were infected by a strain with an increased proportion of naturally resistant mutants, escaping detection despite using advanced molecular techniques. A very low proportion of resistant mutants (between 1/100 and the naturally occurring *rpoB* mutation rate of $1/10^8$) cannot be detected, even by the most sensitive DST, particularly if low-fitness mutations further decrease growth during primary culture and DST. [48]

## Limitations and strengths

Our study has some limitations, particularly the passive follow-up of relapse without systematic follow-up assessment of cured patients, though exceptional efforts were made to capture and register incident relapses as a periodical update of the original records in the individual treatment database. The large majority of studies included in the 2017 meta-analysis equally suffered from this limitation, the effect being likely larger in the observational studies with generally larger participant numbers than in the few included clinical trials. Our dataset actually contains patients enrolled in a clinical trial who had a 12-month post-cure follow-up with complete clinical and bacteriologic assessment.[17] This cohort comprised almost half of our patients with Hr/Rs-TB treated with a 6-month RMP regimen, had DST by targeted Next Generation Sequencing and showed neither failures nor relapses. Moreover, the close similarity of

our findings with these of the original BMRC clinical trials lends additional credibility to our findings.

Another limitation is our inability to distinguish between true relapse and reinfection or true acquired Rr and superinfection with an Rr-TB strain because we could not confirm strain identity by fingerprinting to exclude reinfection. We therefore pragmatically chose to not declare relapse if the recurrent episode was diagnosed more than 2 years after the treatment completion of the previous episode.

We used microscopy-based definitions for failure and relapse. After the introduction of fluorescence microscopy the demonstration of rare AFB in sputum smears increased.[49] In turn, this may have increased the number of patients, falsely declared as treatment failure what actually was late excretion of few residual non-viable bacilli. This may have introduced a bias, as true failure may have been relatively more frequent among cases with initial isoniazid resistance. This would be in line with the greater risk of acquired Rr with Hr reported here.

Contact patients of Rr-TB patients were virtually the only previously untreated cases with DST between the surveys and studies included in the analyses of the 6-month RMP-based regimens. The very small number of Hr/Rs among them may not have biased the 6-month RMP regimen outcome analysis. Since they were exclusively responsible for acquired Rr after this regimen, our reported frequency of acquired resistance for the 6-month RMP-based regimen is likely overestimated.

Finally, social, behavioural and other individual data were not recorded in the clinics or on the treatment cards and could thus also not be captured in the databases. In this Muslim country, alcohol consumption is very rare outside tribal minorities, while cigarette smoking is very frequent among males. HIV is virtually absent among TB patients, as shown by *ad hoc* surveys in the capital but also among the Damien Foundation patients (about 1/1000). Apart from smoking, most likely there has been minimal interference with our findings by these non-quantified factors.

Strengths of our study include the harmonized outcome definitions across cohorts with multiple treatment regimens, and more rigorous phenotypic and genotypic DST that allows robust classification of Hr/Rs versus Hr/Rr.

## Conclusion and recommendations

Our findings contrast with those from the 2017 meta-analysis. [12] In our population initial Hr/Rs was not associated with an excess of adverse bacteriological outcome frequency among patients on the RMP-throughout Category 1 treatment regimen, and only moderately for Category 2 retreatment. Apart from the erroneous inclusion as WHO Category 1 or Category 2 of some large studies with inferior regimens, initial Rr missed by phenotypic DST, particularly in those with previously treated TB and with poly-Hr, may explain the discordance.

Our study confirms that Rr, missed initially or acquired during an RMP-throughout regimen, is by far the most important predictor of first-line recurrence. It accounted for more than half of bacteriologically adverse outcomes among all Rr-TB patients treated with any first-line regimen. To detect both missed initial Rr and acquired Rr we recommend repeating RMP DST after 1 or 2 months treatment in groups at risk, such as previously treated patients, MDR-TB contacts, and those with poly-Hr. Only molecular techniques should be relied upon for ruling out Rr as a condition to prescribe regimens for Hr-TB, even if Rr can still be missed, particularly because of Rr heteroresistance,[42] and because not all *rpoB* resistance mutations are covered.[50]

The 8-month regimen with RMP only in the intensive phase was inadequate for Hr/Rs-TB. Replacing it with RMP-throughout regimens has overcome this inadequacy. However, with

the widespread use of RMP-throughout regimens RMP resistance has evolved into a new global problem that couldn't be overcome until highly effective fluoroquinolone-based regimens had been developed.[51] What we consider an ill-advised large scale use of a fluoroquinolone in a first-line regimen for a wrongly perceived problem will quite possibly increasingly result in the loss of this core drug as a cornerstone of second-line treatment.

## Acknowledgments

We thank the study participants and staff of Damien Foundation Bangladesh.

## Author Contributions

**Conceptualization:** Armand Van Deun, Tom Decroo.

**Data curation:** Armand Van Deun, Aung Kya Jai Maug, Mohamed Anwar Hossain, Murid Gumusboga, Wim Mulders.

**Formal analysis:** Armand Van Deun, Tom Decroo.

**Investigation:** Mohamed Anwar Hossain, Murid Gumusboga, Wim Mulders.

**Methodology:** Armand Van Deun, Tom Decroo, Hans L. Rieder.

**Resources:** Nimer Ortuño-Gutiérrez, Lutgarde Lynen, Bouke C. de Jong.

**Software:** Armand Van Deun, Tom Decroo.

**Supervision:** Armand Van Deun, Hans L. Rieder.

**Validation:** Tom Decroo.

**Writing – original draft:** Armand Van Deun, Tom Decroo, Hans L. Rieder.

**Writing – review & editing:** Aung Kya Jai Maug, Mohamed Anwar Hossain, Murid Gumusboga, Wim Mulders, Nimer Ortuño-Gutiérrez, Lutgarde Lynen, Bouke C. de Jong.

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
