## [Decision Letter · Decision Letter 0]

16 Apr 2020

PONE-D-20-07659

The perceived impact of isoniazid resistance on outcome of first-line rifampicin-throughout regimens is largely due to missed rifampicin resistance

PLOS ONE

Dear Dr. Deun,

Thank you for submitting your manuscript to PLOS ONE. After careful consideration, we feel that it has merit but does not fully meet PLOS ONE’s publication criteria as it currently stands. Therefore, we invite you to submit a revised version of the manuscript that addresses the points raised during the review process.

We would appreciate receiving your revised manuscript by May 31 2020 11:59PM. To enhance the reproducibility of your results, we recommend that if applicable you deposit your laboratory protocols in protocols.io, where a protocol can be assigned its own identifier (DOI) such that it can be cited independently in the future. For instructions see: http://journals.plos.org/plosone/s/submission-guidelines#loc-laboratory-protocols

We look forward to receiving your revised manuscript.

Kind regards,

HASNAIN SEYED EHTESHAM

Academic Editor

PLOS ONE

Journal Requirements:

3. Your ethics statement must appear in the Methods section of your manuscript. If your ethics statement is written in any section besides the Methods, please move it to the Methods section and delete it from any other section. Please also ensure that your ethics statement is included in your manuscript, as the ethics section of your online submission will not be published alongside your manuscript.

Additional Editor Comments (if provided):

Minor Revision

Reviewers' comments:

Reviewer's Responses to Questions

**Comments to the Author**

1. Is the manuscript technically sound, and do the data support the conclusions?

Reviewer #1: Yes

Reviewer #2: Yes

2. Has the statistical analysis been performed appropriately and rigorously? 

Reviewer #1: Yes

Reviewer #2: Yes

3. Have the authors made all data underlying the findings in their manuscript fully available?

Reviewer #1: Yes

Reviewer #2: Yes

4. Is the manuscript presented in an intelligible fashion and written in standard English?

Reviewer #1: Yes

Reviewer #2: Yes

5. Review Comments to the Author

Reviewer #1: Comments to the authors

The study by Armand Van Deun et al. studied a cohort of 7291 patients from Bangladesh for the prediction of various tuberculosis resistance, mainly the involvement of initial isoniazid resistance, rifampicin-resistant, as well as multi-drug resistance in the success of various treatment regimens. The authors mentioned the success of throughout rifampicin resistance in the treatment of isoniazid-resistant TB as well as a failure of this regimen in the treatment of missed rifampicin and acquired rifampicin-resistant TB cases. Importantly, the authors mentioned that failure of other regimens, including throughout rifampicin treatment in the initially missed cases of rifampicin-resistant or acquired resistance to rifampicin, is mainly due to the detection limit in the determination of rifampicin resistance. It is evidenced from the published literature that Rifampicin resistance is commonly used as a predictor of multi-drug resistance in M. tb, also confirmed by the analysis of this study. The study is well conducted; data is well organized and presented. The manuscript is well written and understandable except the abstract that can be written in a more straightforward way to be useful for the wider audience

Abstract:

The abstract written is quite complicated. It is hard to understand. Authors are requested to write an abstract that can easily be understood.

Reviewer #2: Comments:

Overall the manuscript is broadly written about the treatment categories related to active TB. Data proved the treatment utility and recommendations in present scenario for TB programme successfully progresses worldwide. Following are some suggestions which can make the manuscript stronger and beneficial for the readers:

Abstract:

• Abbreviation of rifampicin resistance (Rr) should come upon first appearance in the text.

• Conclusive statement should be modified to give good interpretation to the reader.

Introduction:

• Abbreviation must require for rifampicin (RMP) upon first appearance in the text (Line no 60).

Method: It observed that author/s ware collected the data very precisely. However, required some information to improve manuscript like:

• Author should point out the type (pulmonary or extra-pulmonary) of tuberculosis among patient requited for this study as well as house hold contact, how developed active TB.

• In patient history, alcoholism and smoker status of patient as well as house hold contact should be noted if possible. Because these habits are strongly associated with TB disease and attributed to high risk of TB infection which leads to active TB or recurrence.

Results:

• Authors should provide the data of TB-HIV co-infected patients. Because, they may have negative in sputum smear microscopy and required initial TB treatment.

6. PLOS authors have the option to publish the peer review history of their article (what does this mean?). If published, this will include your full peer review and any attached files.

Reviewer #1: Yes: Mohd Shariq

Reviewer #2: No

---

## [Author Response · Author response to Decision Letter 0]

4 May 2020

PONE-D-20-07659

The perceived impact of isoniazid resistance on outcome of first-line rifampicin-throughout regimens is largely due to missed rifampicin resistance

Response to the reviewers

Reviewer #1: Comments to the authors

The study by Armand Van Deun et al. studied a cohort of 7291 patients from Bangladesh for the prediction of various tuberculosis resistance, mainly the involvement of initial isoniazid resistance, rifampicin-resistant, as well as multi-drug resistance in the success of various treatment regimens. The authors mentioned the success of throughout rifampicin resistance in the treatment of isoniazid-resistant TB as well as a failure of this regimen in the treatment of missed rifampicin and acquired rifampicin-resistant TB cases. Importantly, the authors mentioned that failure of other regimens, including throughout rifampicin treatment in the initially missed cases of rifampicin-resistant or acquired resistance to rifampicin, is mainly due to the detection limit in the determination of rifampicin resistance. It is evidenced from the published literature that Rifampicin resistance is commonly used as a predictor of multi-drug resistance in M. tb, also confirmed by the analysis of this study. The study is well conducted; data is well organized and presented. The manuscript is well written and understandable except the abstract that can be written in a more straightforward way to be useful for the wider audience

Abstract:

The abstract written is quite complicated. It is hard to understand. Authors are requested to write an abstract that can easily be understood.

 Thank you so much for this important remark. We have revised the abstract extensively so as to make it easily understandable.

Reviewer #2: Comments:

Overall the manuscript is broadly written about the treatment categories related to active TB. Data proved the treatment utility and recommendations in present scenario for TB programme successfully progresses worldwide. Following are some suggestions which can make the manuscript stronger and beneficial for the readers:

Abstract:

• Abbreviation of rifampicin resistance (Rr) should come upon first appearance in the text.

We are sorry for this lapse, it has been corrected.

• Conclusive statement should be modified to give good interpretation to the reader.

The abstract has been revised entirely, including the conclusion, as advised by Reviewer 1. Moreover, since we stress the danger of acquired fluoroquinolone resistance as more important than the negative impact of isoniazid resistance, we thought this needed some more explanation. We have added a few sentences to explain why acquired resistance could come up quite readily and frequently in the discussion about the strengthened regimen.

Introduction:

• Abbreviation must require for rifampicin (RMP) upon first appearance in the text (Line no 60).

We are sorry for this lapse, it has been corrected.

Method: It observed that author/s ware collected the data very precisely. However, required some information to improve manuscript like:

• Author should point out the type (pulmonary or extra-pulmonary) of tuberculosis among patient requited for this study as well as house hold contact, how developed active TB.

Thank you. All patients were pulmonary cases, and this is now specified in the manuscript. Unfortunately contacts and secondary cases that developed from them were as a rule not registered in the computer database, although they were listed on the original paper treatment cards that we couldn't analyse. This may not be a serious deficiency because outside the systematic population surveys, in fact only contacts of MDR cases were tested for baseline drug resistance. This contact status has been recorded in the database and we have now added their number included in the study in the text (methods as well as results). 

• In patient history, alcoholism and smoker status of patient as well as house hold contact should be noted if possible. Because these habits are strongly associated with TB disease and attributed to high risk of TB infection which leads to active TB or recurrence.

We are sorry, but much of this information is not available because it all started as an observational study, with close to 300,000 cases individually entered into the original patient database, focusing on the variables considered to be most important for analyses of diagnostic and treatment parameters only, not risk factors for the disease or recurrence. Smoking, occupation and household contact were thus not part of the computerised database variables and therefore unavailable for analysis. However, we can confidently state that alcoholism exists in this largely Muslim country only among tribal non-Muslim minorities, very few of whom are included in our dataset. While smoking is very common among males. 

Results:

• Authors should provide the data of TB-HIV co-infected patients. Because, they may have negative in sputum smear microscopy and required initial TB treatment.

We apologise again, because HIV status in Bangladeshi TB patients has been determined only in the context of ad hoc studies. That consistently showed a very low prevalence of the order of 1 per 1000 patients, explaining that routine testing of all patients was not considered cost-effective by the TB control programme authorities. We have added this fact, and that HIV thus cannot have influenced our findings.

---

## [Editor Report · Decision Letter 1]

7 May 2020

The perceived impact of isoniazid resistance on outcome of first-line rifampicin-throughout regimens is largely due to missed rifampicin resistance

PONE-D-20-07659R1

Dear Dr. Deun, 

We are pleased to inform you that your manuscript has been judged scientifically suitable for publication and will be formally accepted for publication once it complies with all outstanding technical requirements.

With kind regards,

HASNAIN SEYED EHTESHAM

Academic Editor

PLOS ONE

Additional Editor Comments (optional):

In this manuscript the Authors studied a cohort of patients from Bangladesh for the prediction of various tuberculosis resistance. The reviewers made important suggestions and had questions which the authors have addressed satisfactorily. Abstract of the manuscript has been revised extensively to make it easily understandable. The inability to provide the data on TB-HIV co-infected patients is accepted. I recommend publication of this manuscript.
---

## [Editor Report · Acceptance letter]

8 May 2020

PONE-D-20-07659R1 

The perceived impact of isoniazid resistance on outcome of first-line rifampicin-throughout regimens is largely due to missed rifampicin resistance 

Dear Dr. Van Deun:

I am pleased to inform you that your manuscript has been deemed suitable for publication in PLOS ONE. Congratulations! Your manuscript is now with our production department. 

With kind regards,

on behalf of

Prof HASNAIN SEYED EHTESHAM 

Academic Editor

PLOS ONE